# Mixed Matrix Membranes for Efficient CO_2_ Separation Using an Engineered UiO-66 MOF in a Pebax Polymer

**DOI:** 10.3390/polym14040655

**Published:** 2022-02-09

**Authors:** Asmaul Husna, Iqubal Hossain, Insu Jeong, Tae-Hyun Kim

**Affiliations:** 1Organic Material Synthesis Laboratory, Department of Chemistry, Incheon National University, Incheon 22012, Korea; shopnilchem@inu.ac.kr (A.H.); jis_3088@naver.com (I.J.); 2Research Institute of Basic Sciences, Incheon National University, Incheon 22012, Korea

**Keywords:** gas separation membrane, Pebax matrix, interface manipulation, PIM-grafted-MOF, anti-aging performance

## Abstract

Mixed matrix membranes (MMMs) have attracted significant attention for overcoming the limitations of traditional polymeric membranes for gas separation through the improvement of both permeability and selectivity. However, the development of defect-free MMMs remains challenging due to the poor compatibility of the metal–organic framework (MOF) with the polymer matrix. Thus, we report a surface-modification strategy for a MOF through grafting of a polymer with intrinsic microporosity onto the surface of UiO-66-NH_2_. This method allows us to engineer the MOF–polymer interface in the MMMs using Pebax as a support. The insertion of a PIM structure onto the surface of UiO-66-NH_2_ provides additional molecular transport channels and enhances the CO_2_ transport by increasing the compatibility between the polymer and fillers for efficient gas separation. As a result, MMM with 1 wt% loading of PIM-grafted-MOF (PIM-g-MOF) exhibited very promising separation performance, with CO_2_ permeability of 247 Barrer and CO_2_/N_2_ selectivity of 56.1, which lies on the 2008 Robeson upper bound. Moreover, this MMM has excellent anti-aging properties for up to 240 days and improved mechanical properties (yield stress of 16.08 MPa, Young’s modulus of 1.61 GPa, and 596.5% elongation at break).

## 1. Introduction

Greenhouse gas production is increasingly posing a serious environmental threat, and, as a result, CO_2_ separation technologies have attracted significant attention as a potential way to mitigate rising CO_2_ levels in the atmosphere [1,2,3,4]. Among the various CO_2_ separation techniques, membrane separation stands out for its low energy consumption, environmental friendliness, simple operation, ease of maintenance, and good performance [4,5,6,7]. Membranes are widely used to separate CO_2_ from different mixtures, such as fossilfuel combustion gases, natural gas, and synthetic gas [6,8,9].

In general, the organic polymer membranes with the best gas separation performance are considered to be those with high CO_2_ permeability combined with high CO_2_ selectivity. However, there is a general trade-off relationship between the permeability *P* (the product of diffusivity and solubility, *P* = *D* × *S*) and selectivity α (the permeability ratio between two gases, *P_A_*/*P_B_*). More permeable polymers are generally less selective, which is defined by the Robeson upper bound [10,11].

With this relationship in mind, various methods have been proposed to improve the separation performance of polymeric membranes, including polymer blending, crosslinking, copolymerization, and the incorporation of fillers [6,7,12,13]. Incorporating nonporous or porous fillers into a polymer matrix leads to the formation of mixed matrix membranes (MMMs), which have great potential as high-performance membranes. MMMs are hybrids between a polymer (as the continuous phase or matrix) and inorganic nanoparticles (as the dispersed phase or additive). Thus, they offer the combined properties of both polymers, such as mechanical flexibility, facile processing, low cost, and easy upscaling, and those of inorganic fillers, such as excellent thermo-mechanical stability, high free volume, and selectivity [1,14,15]. Accordingly, these membranes offer the possibility of breaking through the Robeson upper bound of typical organic polymer membranes used for gas separation or even avoiding the trade-off relationship entirely [1,14,16]. However, the choice of both the polymer and filler is critical, as these characteristics affect the morphology and, hence, the resulting gas separation performance of MMMs.

Various porous or nonporous materials have been incorporated as fillers in MMMs, including zeolite, metal oxides, mesoporous silica, graphene, graphene oxide, and metal–organic frameworks (MOFs) [17,18,19,20,21,22]. MOFs in particular have gained great attention for their high BET surface area, porous and rigid network structure, and structure-tuning ability [23,24,25]. Several MOFs have been used to fabricate MMMs, such as Cu_3_BTC_2_ [26]_,_ ZIF-8 [27], MOF-5 [28], MIL-53 [29], and HKUST-1 [30].

UiO-66-NH_2_ is an amine-functionalized zirconium-carboxylate MOF that has been used to prepare MMMs. It has been studied for its high thermochemical and hydrolytic stability, which originate from the strong coordination bonds between hard-acid Zr(IV) atoms and the hard-base carboxylate oxygen atom [31,32,33,34]. In addition, the amine end-group is highly reactive and allows for further modifications [35,36,37]. Various UiO-66-NH_2_-based MMMs have been developed which have proven their potential as efficient CO_2_-separation membranes [13,38].

In our previous studies, we developed a new MMM by combining PIM-PI-functionalized UiO-66 MOF (PPM) with a PIM-PI-1 polymer matrix for effective interactions between the MOF and the polymer matrix, which resulted in excellent CO_2_ separation performance [39]. However, that approach has some definite limitations for large-scale applications. First, only a small fraction of the MOF surface was modified, due to the incorporation of the long chains (oligomers) of PIM-PI. Second, the multi-step oligomer synthesis reaction used extremely expensive starting materials. Moreover, although the short-term anti-aging performance was improved, some aging still occurred due to the high free volume of the polymer. The selectivity was also not high enough, due to the highly microporous structure of PIM-PI-1. These limitations need to be resolved for large-scale industrial applications.

We used Pebax as a support layer in the present study, because it is well known as a high-performance thermoplastic elastomer matrix with good mechanical strength as well as excellent separation and adhesion properties. These properties stem from the micro phase-separated structure containing polyamide (PA) and polyethylene oxide (PEO). Furthermore, Pebax has very strong interactions between the filler and the matrix, which often significantly reduces CO_2_ permeability.

To enhance the gas separation performance and mechanical properties of the new membrane, PIM-g-MOF was fabricated by grafting polymerization using a polymer of intrinsic microporosity on the UiO-66-NH_2_ MOF surface. It was then incorporated into the Pebax matrix to fabricate the PIM-g-MOF-x MMMs. The PIM-g-MOFs are expected to have better dispersibility in the Pebax matrix due to the functionalization of the MOF particles. The insertion structure on the surface of the UiO-66-NH_2_ MOF enhances the organic part of the particles, which is expected to prevent them from aggregating.

Additionally, the PIM has high CO_2_ gas permeability compared with other polymers, leading to better gas transfer. The polymerized PIM particles extending from the surface of the MOF particles could be used as transport channels for gas molecules in the polymer matrix and, thus, effectively mitigate the adverse effect of the Pebax matrix on the CO_2_ permeability. The effects of the content of the newly developed fillers on the gas transport property are reported, and the effects on thermal stability, morphology, anti-aging, anti-plasticization, and temperature were also examined.

We developed mixed matrix membranes incorporating PIM-grafted-MOF (PIM-g-MOF) (**1**) using Pebax as a support polymer in the present study. We used 4,5-dichloro phthalic (**4**) anhydride as a cheap starting material to modify the MOF by grafting with PIM-PI polymer. Thus, direct polymer grafting occurred on the MOF surface for MOF modification, thereby increasing the amount of organics in the MOF structure. Furthermore, the final structure of the PIM-g-MOF (Figure 1) is different from our previous PPM MOF: the durene monomer (diamine) was used for the polymerization, but in the present study the functional group of UiO-66-NH_2_ (Figure 1) was used as a reactive monomer for the polycondensation reaction. The dichloro monomer also acts as a repeating unit in the grafting polymerization instead of durene. The organic functionality in the newly developed MOF structure is, hence, much higher than in our previous report. The technique presented is an in situ MOF modification approach that can be used for direct in situ polymerization of MOFs for the development of MMMs.

## 2. Materials and Methods

### 2.1. Materials

The monomer4,5-dichlorophthalic anhydride was purchased from Tokyo Chemical Industry (TCI) Co. Ltd. (Tokyo, Japan); 5,5′,6,6′-tetrahydroxy-3,3,3,3′-tetramethyl-1,1′-spirobisindane (97%) was purchased from Alfa Aesar (Seoul, Korea). These were used as obtained. Zirconium chloride (ZrCl_4_) and 2-aminoterephthalic acid were obtained from Alfa Aesar (Seoul, Korea). Pebax was purchased from ArkemaInc. Methanol, ethanol, chloroform, dimethyl formamide, and potassium carbonate were purchased from DaeJung Chemicals & Metals Co. Ltd. (Shiheung, Korea) in South Korea. Acetic anhydride, acetic acid, and triethylamine were obtained from Sigma Aldrich. (Yongin, Korea)

### 2.2. Preparation of PIM-Grafted-MOF ***1***

UiO-66-NH_2_ (**5**) was prepared according to our previous reported method [13,39]. Briefly, a mixture of 2-aminoterephthalic acid (1.0 g, 5.5 mmol), ZrCl_4_ (1.28 g, 5.5 mmol), and 3.15 mL of acetic acid was dissolved in 80 mL of DMF. The solution was transferred into a Teflon-lined autoclave and then heated in an oven at 120 °C for 24 h. The resulting yellow suspension was centrifuged and washed with methanol several times, redispersed in methanol for one week, and, finally, centrifuged before being activated in a vacuum oven at 120 °C for 24 h.

After the preparation of UiO-66-NH_2_ (**5**) MOF, a two-step synthesis procedure was followed for the preparation of PIM-g-MOF (**1**).

At the first step, 220 mg of UiO-66-NH_2_ MOF was dispersed in 7 mL of DMF with 240 mg of 4,5-dichlorophthalic anhydride, and acetic anhydride (Ac_2_O) and triethyl amine (TEA) were added to the reaction mixture. After that, this mixture was refluxed at 110 °C for 8 h in a round-bottom flask with stirring. The obtained powders were isolated by centrifuge using DMF and chloroform several times to remove any unreacted monomers.

At the second step, after drying, the light yellow powder at 80 °C was further reacted with TTSBI (5,5′,6,6′-tetrahydroxy-3,3,3,3′-tetramethyl-1,1′-spirobisindane (97%)) monomer for grafting the UiO-66-NH_2_ MOF surface with PIM monomer (kink structure). Finally, PIM-g-MOF was synthesized by dispersing 500 mg of dichloro anhydride containing UiO-66-NH_2_ MOF in 15 mL of DMF with tetraol monomer and 3 g of potassium carbonate (K_2_CO_3_). The mixture was refluxed at 120 °C for 24 h in a round-bottom flask with magnetic stirring. The obtained powders were again washed with DMF and chloroform several times and dried under 80 °C for 2 days. For further use, the centrifugate brown colored powder was stored. Attenuated total reflectance−Fourier transform infrared spectroscopy (ATR−FTIR) (cm^−1^): 2955, 2927, and 2855 (C–H asymmetric and symmetric stretching originating from tetraol monomer); 1735 (imide C=O stretching); 1567 (COO–linked with Zr4+); 1356 (imide C–N symmetric stretching); and 1092 (C–O–C stretching).

### 2.3. Preparation of PIM-g-MOF (***1***)-Incorporated Pebax-Based MMMs, PIM-g-MOF-x

First, 500 mg of Pebax pellets was dissolved in 10 mL of ethanol:water (70:30) mixture with stirring at 80 °C for 3 h. The fillers were dispersed into another 5 mL of ethanol:water mixture (70:30) with ultrasonication and stirring. Then, the dispersed fillers solution was added to the Pebax solution and sonicated for good dispersion of MOF into the matrix. Afterwards, the mixture was poured into a Petri dish and dried at room temperature for 24 h. Finally, the prepared membranes were further dried at 80 °C in a vacuum oven to remove the solvent completely. ATR–FTIR (cm^−1^) for PIM-g-MOF-1: 3298 (N–H stretching); 2945, 2903, and 2862 (C–H asymmetric and symmetric stretching); 1739 (imide C=O stretching); 1637 (H–N–C=O stretching); 1580 (COO–linked with Zr4+); 1347 (imide C–N stretching); 1257 (CH_2_–O–CH_2_ asymmetric stretching); 1097 (CH_2_–O–CH_2_ symmetric stretching); and 950 (CH=CH bending). All other membranes showed similar IR spectra with reduced peak intensities, whereas peak positions for N–H stretching were shifted to lower wavenumbers and, for C=O stretching, were shifted slightly to higher wavenumbers.

### 2.4. Characterization and Measurements

Structures and physical properties were analyzed by ^1^H NMR and ATR–FTIR spectroscopies. Thermal properties were analyzed by thermogravimetric analysis (TGA), N_2_ adsorption experiments were conducted at 77 K using a Micromeritics ASAP 2020 HD88, and mechanical properties were measured by a universal testing machine (UTM). The morphology of the membranes was determined using wide-angle X-ray diffraction (WAXD), scanning electron microscopy (SEM), and atomic force microscopy (AFM). The gas separation performance was measured by using the time-lag instrument under constant volume–variable pressure conditions. Details on instrumentation and characterization are available in the Appendix A.

## 3. Results and Discussion

### 3.1. Synthesis and Characterization of UiO-66-NH_2_(***5***) and Its PIM-Grafted-MOF Form, PIM-g-MOF (***1***)

UiO-66-NH_2_ (**5**) was synthesized by reacting zirconium chloride and 2-aminoterephthalic acid according to a previously reported procedure (Appendix A) [13,39,40]. UiO-66-NH_2_ (**5**) was further reacted with 4,5-dichlorophthalic anhydride (**4**) to obtain the imide MOF (**2**) comonomer with dichloro functional groups (active sites for polymerization) and polymerized with the monomer TTSBI (5,5′,6,6′-tetrahydroxy-3,3,3′,3′-tetramethyl-1,1′-spirobisindane) (**3**) through a polycondensation reaction to form the PIM-g-MOF (**1**) (Figure 1).

The FTIR spectra of the PIM-g-MOF (**1**) showed specific changes in the absorption peaks compared to that of UiO-66-NH_2_ (**5**). The characteristic peaks for UiO-66-NH_2_ (**5**) are at 3469 cm^−1^ (N–H asymmetric stretching), 3361 cm^−1^ (N–H symmetric stretching), 1662 cm^−1^ (–NH_2_ bending), and 1100 cm^−1^ (C–NH_2_ stretching). These peaks decreased significantly for the functionalized PIM-g-MOF (Figure 2a).

In addition, the spectrum of the PIM-g-MOF (**1**) showed new peaks centered at 1716 cm^−1^ (stretching of C=O), 1358 cm^−1^ (imide C–N stretching), and 2935 cm^−1^ and 2853 cm^−1^ (symmetric and asymmetric C–H stretching) originating from the PIM-PI polymer. These results support the case for the successful grafting of UiO-66-NH_2_ to obtain the corresponding PIM functionality (i.e., PIM-g-MOF, **1**). Nevertheless, there is a very small peak at 1656 cm^−1^ (N–H bending) and a broad peak centered at 3364 cm^−1^ (N–H stretching), indicating that some of the amine groups on the MOF structure (UiO-66-NH_2_) were not completely polymerized with PIM functionality and remained in their unmodified amine forms.

The ^1^H NMR spectra of the UiO-66-NH_2_ (**5**) and PIM-g-MOF (**1**) were also analyzed by digesting the PIM-g-MOF using NaOD (aq) solution, followed by sonication [41]. There were new peaks at 7.36–7.31 ppm and 6.85–6.80 ppm, which were attributed to ArH, while those at 1.9 ppm and 1.54 ppm were attributed to CH and CH_3_ from the polymerized PIM-PI unit of the PIM-g-MOF (**1**) (Figure 2c). Comparison of this spectrum with that of UiO-66-NH_2_ (**5**) (Figure 2b) indicated successfulpolymerization of the amine group in UiO-66-NH_2_ to the desired PIM-g-MOF (Figure 2c).

Appendix A shows scanning electron microscopy (SEM) images of the pristine UiO-66-NH_2_ (**5**) and the PIM-g-MOF. The PIM-g-MOF (**1**) displayed octahedral morphology with an average particle size of approximately 248 ± 20 nm. The pristine MOF UiO-66-NH_2_ (**5**) had the same octahedral morphology, but the average particle size was 141 ±15 nm. These measurements demonstrate that the particle sizes of PIM-g-MOF increased with no changes in the particle morphology compared to UiO-66-NH_2_. This also indicates the successful functionalization of UiO-66-NH_2_ by grafting. In addition, wide-angle X-ray diffraction (WAXD) results showed that the crystal diffraction patterns for both UiO-66-NH_2_ (**5**) and PIM-g-MOF (**1**) were retained after modification, suggesting that there was no or a negligible effect on the crystal structure after the functionalization of the MOFs (UiO-66-NH_2_) with a PIM unit (Appendix A).

The thermal stability of the UiO-66-NH_2_ (**5**) and PIM-g-MOF (**1**) was characterized by thermal gravimetric analysis (TGA, Appendix A). The first weight loss occurred around 50–150 °C for UiO-66-NH_2_ due to the release of moisture [42]. The second weightloss step occurred around 200–250 °C due to the decomposition of DMF from the pores. The third weightloss step was observed at higher temperatures and was ascribed to the decomposition of the organic linker and framework.

In the PIM-g-MOF (**1**), the weight loss decomposition above 300 °C was lower than the weight loss decomposition of UiO-66-NH_2_ and is attributed to the likely thermal degradation of the PIM chains attached to the organic linker of the MOF structure. In contrast, the thermal degradation of UiO-66-NH2 begins near 265 °C, suggesting that the thermal stability of the PIM-g-MOF (**1**) increased due to the higher thermal stability of PIM-PI compared to the MOF. This is another indication of the successful functionalization of the MOF.

The nitrogen adsorption/desorption behavior of UiO-66-NH_2_ (**5**), imide MOF (**2**), and PIM-g-MOF (**1**) was measured using BET at 77 K (Appendix A). The Brunauer–Emmett–Teller (BET) surface areas of **1**, **2**, and **5** were 1041.8 m^2^ g^−1^, 801.5 m^2^ g^−1^, and 141.8 m^2^ g^−1^, respectively, indicating the surface area of the PIM-g-MOF was reduced dramatically after polymerization compared to the surface area of UiO-66-NH_2_. It also suggests that the free volume was occupied by PIM units on the surface of the MOF. The apparent loss of surface area caused by a partial blockage of the MOF pores by the guest molecules has also been reported elsewhere [13,25,37,39,40].

Nevertheless, the isotherm curves of three samples revealed that all of the MOFs displayed microporosity: the isotherm of PIM-g-MOF showed a hysteresis loop similar to that of the pristine PIM-PI-1 polymer [4,39], which was absent in UiO-66-NH_2_. The overall results demonstrate the successful polymerization of the MOF (UiO-66-NH_2_) with PIM units to form PIM-g-MOF.

### 3.2. Synthesis of PIM-g-MOF-x@Pebax

A series of MMMs was prepared with different PIM-g-MOF loadings using ethanol:water (3:1) and the solvent casting method. First, Pebax was dissolved in a solvent mixture of ethanol and water; PIM-g-MOF (**1**) was dispersed in the same solvent in two different vials, and the mixtures were sonicated for 20 min. Then, the filtered Pebax solution was added to the MOF suspension and mixed for 2 h. This multicomponent suspension was used to prepare self-standing membranes using a slow solvent evaporation method after casting on a Petri dish. The membrane was peeled off the dish and dried in an 80 °C oven for 72 h under high vacuum. All loadings formed uniform and flexible films (Appendix A).

As-prepared MMMs from PIM-g-MOF (**1**) and Pebax are simply designated as PIM-g-MOF-x, where x represents the percentage of MOF loading in the Pebax support. Four different MMMs were prepared using 0.5, 1, 3, and 5 wt% MOF: PIM-g-MOF-0.5, PIM-g-MOF-1, PIM-g-MOF-3, and PIM-g-MOF-5, respectively. We also prepared a blended membrane for comparison by mixing different loadings of the pristine UiO-66-NH_2_ (**5**) with the Pebax copolymer. This was performed using the same method used for the preparation of PIM-g-MOF-x for in situ membrane casting. This was carried out to investigate the effect of the chemically functionalized UiO-66 on the properties of the corresponding PIM-g-MOF-x. These membranes are designated as UiO-66-NH_2_-1 and UiO-66-NH_2_-3 for MMMs with 1 and 3 wt% pristine UiO-66-NH_2_ MOF loading, respectively.

### 3.3. Characterization of the PIM-g-MOF-x MMMs

The structure of the PIM-g-MOF-x MMMs with different MOF loadings was characterized by comparative spectroscopic analysis using ATR–FTIR. IR spectra of the precursor MOF (PIM-g-MOF, **1**) and the pristine Pebax polymer (Appendix A) were also included for comparison. ATR–FTIR spectroscopy was used to study the interaction between the fillers and the Pebax polymer matrix. The characteristic peaks for the N–H stretching (at 3303 cm^−1^), H–N–C=O stretching (at 1640 cm^−1^), and C–O–C stretching (at 1100 cm^−1^) are assigned to the Pebax polymer backbone (relating to the amide bonds and PEG chain). In addition, the intensity and peak area of the signals at 2945 and 2865 cm^−1^ corresponding to the symmetric and asymmetric C–H stretching vibrations, respectively, were also observed for the Pebax polymer (Appendix A).

Some of the characteristic peaks (C=O, C–O–C) of the Pebax membrane were reduced in intensity and shifted slightly toward higher frequencies (Appendix A), due to the presence of the PIM-g-MOF (**1**) particles. This suggests that incorporating the PIM-g-MOF (**1**) particles in the polymer matrix partially breaks the hydrogen bonds of the amide nitrogen of the PA segments with each other or with the carbonyl oxygens of other chains, which are associated with the polymer chains and the induced microcrystalline structure [43]. Therefore, as shown in Appendix A, the FTIR spectra of the MMMs are very similar to that of the pristine Pebax membrane, including the N–H stretching vibration of the amide group, which consists of two peaks at 3309 and 3205 cm^−1^. These peaks are assigned to the free and hydrogen-bonded N–H groups, respectively.

In the presence of PIM-g-MOF (**1**) particles, a considerable reduction in intensities was observed for both peaks, and the hydrogen-bonded peak shifted from 3205 cm^−1^ to 3194 cm^−1^, due to the partial disruption of these hydrogen bonds. The peak attributed to the hydrogen-bonded stretching vibration of the carbonyl group, H–N–C=O, declined in intensity and moved from 1645 cm^−1^ to 1635 cm^−1^ for the same reason. Furthermore, the intensity was also reduced for the peak assigned to the stretching vibration of the ether group at 1101 cm^−1^. This supports the formation of a hydrogen bond between the amide bond (–NH group) of the Pebax polymer and the imide carbonyl (C=O) of the MOF. Thus, the presence of the MOF inhibits the free movement of some functional groups in the Pebax, due to the hardening effect of the fillers [44]. Therefore, we concluded that the embedded PIM-g-MOF (**1**) was successfully incorporated into the Pebax matrix.

### 3.4. Thermal and Mechanical Properties of MMMs

The thermal stability of the newly developed MMMs was investigated by thermogravimetric analysis (TGA) and compared with those of PIM-g-MOF (**1**) (Figure 3c,d). The neat Pebax membrane had only one thermal weightloss stage in the range of 350–490 °C, which was ascribed to the thermal degradation of the molecular chains of Pebax [44]. In general, the shape of the thermal weight loss curve of the MMMs is similar to that of neat Pebax, with one major thermal decomposition stage centered around 360 °C.

As the PIM-g-MOF (**1**) content increased, the onset temperature of the major pyrolysis of the MMMs decreased from 350 to 338 °C. This behavior is attributed to the reduction of intermolecular cohesive energy because of the obstruction of more hydrogen bonds by adding more PIM-g-MOF (**1**). At the same time, although the thermal weight loss trend of the MMMs slowed down in the range of 440 to 600 °C, the maximum weight loss temperatures (*T_max_*) were similar for all PIM-g-MOF-x, as shown in Appendix A. These results are caused by the interference of added PIM-g-MOFs in the membrane matrix, but the MMMs retained good thermal stability at high temperatures.

The mechanical properties of the membranes were measured using UTM to investigate the effect of PIM-g-MOF (**1**) fillers on the tensile stress, elongation, and Young’s modulus of the membranes, as shown in Figure 3c,d and Appendix A. The interaction of the fillers with polymer segments has a major effect on the plasticity, tensile strength, and elasticity of the entire membrane. The Young’s modulus increased gradually with increasing PIM-g-MOF (**1**) filler loading in the MMMs from 0.80 GPa to 1.61 and 2.01 GPa, as shown in Appendix A. This confirms that there are stronger interfacial interactions between the polymer and the filler. Unfortunately, the Young’s moduli of the membranes experienced a loss at high filler contents (5 wt%) due to increasing filler agglomeration, which deteriorates the mechanical properties of the membranes. However, with high filler content, PIM-g-MOF-5 still shows a higher Young’s modulus (142% more) than the pristine Pebax membrane.

Furthermore, the tensile strength of the MMMs also increases with the increase of MOF loading, indicating improved interaction between the MOF and the polymer matrix. In contrast, the elongation at break of the MMMs decreases with the addition of PIM-g-MOF (**1**), which could be explained by the reduced membrane ductility induced by polymer chain rigidification at the polymer–filler interfaces [45]. Therefore, the addition of PIM-g-MOF (**1**) can effectively improve the mechanical properties of the pristine Pebax membrane by enhancing the interaction between the polymer and MOF, and the MMMs have sufficient stability for use as gas separation membranes.

### 3.5. Morphological Analyses by WAXD, SEM, and AFM

The XRD patterns of Pebax and MMMs (Figure 4) obtained using WAXD were used to study the arrangement of the fillers and the structural changes of the polymer chains. The characteristic peaks of the polyamide crystalline phase and the non-crystalline PEO segment of the Pebax polymer were between 2θ = 15–24°. In Figure 4, new peaks appear for the MMMs around 2θ = 6.5°, 11.4°, 20.1°, and 23.1°, which correspond to the PIM-g-MOF (**1**) structure. With increased content of PIM-g-MOF (**1**) filler, the MOF peaks become more intense, indicating more filler loading in the MMMs. Moreover, the polymer peak pattern became marginally broader, indicating that the PIM-g-MOFs disturb the chain packing of the Pebax polymer, thereby creating free volume for gas transport.

The microstructure of the PIM-g-MOF-incorporated MMMs (PIM-g-MOF-x) was further analyzed to investigate the dispersion and interfacial adhesion by studying the surface and cross-section morphologies of the MMMs using FE-SEM (Figure 5 and Appendix A). The results were compared to the SEM analysis result of the pristine polymer membrane (Pebax) and those of the unmodified UiO-66-NH_2_-loaded blended membranes (UiO-66-NH_2_-1 and UiO-66-NH_2_-3) (Appendix A). The pristine polymer membrane showed a smooth and defect-free topography (Figure 5a). However, the blended membranes of UiO-66-NH_2_-1 (Appendix A) and UiO-66-NH_2_-3 (Appendix A) displayed an aggregated internal topography, which occurred at the interface between the Pebax polymer and the UiO-66-NH_2_ fillers, due to the lack of effective MOF–polymer interactions. The particle aggregation shown in Appendix A could potentially have negative effects on the gas separation performance.

In contrast, the surface and cross-section images of all of the MMMs (PIM-g-MOF-x) showed defect-free microstructures, although the distribution of the MOF on the Pebax polymer depended on the MOF loading (Figure 5b,c and Appendix A). The SEM images of the MMMs showed good dispersion of PIM-g-MOF for PIM-g-MOF-0.5 and PIM-g-MOF-1 (Figure 5b,c and Appendix A). However, the MOF particles began to aggregate slightly at PIM-g-MOF-3 (Figure 5d and Appendix A), and aggregation was more pronounced at 5% loading in PIM-g-MOF-5 (Figure 5e and Appendix A).

A uniform dispersion of MOFs allows the transport of gas molecules through the porous structures of the MOF via rapid diffusion, which leads to increased gas permeability. In contrast, a particle-aggregated morphology of the membrane leads to lower gas separation performance due to the blocking of pores or the densification of polymers surrounding the MOF–polymer interface. Therefore, despite the formation of uniform and defect-free membranes for all MMMs with various MOF loadings, the morphological analyses of the MMMs by SEM suggest that there is an optimum level of MOF loading to obtain the most effective gas separation performance with a well-dispersed MOF morphology. In addition, the PIM-g-MOF-x MMMs showed better particle dispersion in the membrane than previous reports of MMMs containing UiO-66-NH_2_ MOFs using Pebax as a matrix [46].

The morphology of PIM-g-MOF-x and Pebax in membranes was further investigated by atomic force microscopy (AFM). AFM was used to investigate the changes in the surface morphology and the roughness of the membranes. As shown in Figure 6, the Pebax membrane consists of randomly distributed soft and hard segments, which create an incomplete phase-separated morphology. Pebax is a segmented copolymer with soft rubbery and hard glassy phases that can lead to microphase separation. However, previous observations indicated that the hard glassy PA segment of pristine Pebax has structures of rod-like lamellae that appear brighter in color embedded in the rubbery PEO segment, which appears darker in color in AFM images (Figure 6a) [47].

In comparison, for PIM-g-MOF-1 (Figure 6b), the PA phase near the MOF surface is more organized and largely oriented toward the long axis of the MOFs with a crisscross arrangement. However, in Figure 6c,d, the pristine Pebax membrane has a smooth surface in 3D topography images, while adding 1 wt% PIM-g-MOF to the MMM changes the roughness of the membrane (Figure 6d). The surface and 3D images indicate that the polymer matrix fully surrounded the MOF, and no bulges were observed. Furthermore, the increased surface roughness gives the membranes higher effective surface areas, which should be beneficial for gas contact and its subsequent dissolution on the polymeric surface [48]. Overall, very good phase interaction was observed between the matrix and the PIM-g-MOF, which further demonstrates the strong MOF–polymer compatibility attained in the resultant PIM-g-MOF-x membranes.

### 3.6. Gas Separation Performance of the Mixed Matrix Membranes

The single gas permeability of the MMMs (PIM-g-MOF-x) with varied loadings of PIM-g-MOF (**1**) was measured using the constant-volume/variable-pressure method at 1 atm and 30 °C (detailed instrumentation is reported in our previous work [13], and procedures are illustrated in the Appendix A). Measurements were carried out in triplicate for each gas on three independently fabricated membranes, and average values are reported. The ideal selectivity (termed selectivity for simplicity) was calculated from the ratio between the high-permeabilitygas and the low-permeability gas. Appendix A and Figure 7 show the results of gas separation for N_2_, CH_4_, and CO_2_ and their perm-selectivities (i.e., PCO_2/_N_2_ and PCO_2/_CH_4_).

The pristine Pebax membrane showed CO_2_ permeability of 141.4 Barrer together with CO_2_/N_2_ and CO_2_/CH_4_ selectivities of 35.3 and 11.7, respectively (Appendix A). As shown in Figure 7a, MMMs’ permeabilities to all gases gradually increased with increased PIM-g-MOF loading in the Pebax polymer matrix up to 1 wt%. However, further increases in the loading to 3 wt% PIM-g-MOF filler leads to a decrease in CO_2_ permeability with reduced CO_2_/N_2_ selectivity. The PIM-g-MOF-1 MMM with 1 wt% loading exhibits an optimal CO_2_/N_2_ separation performance with CO_2_ permeability of 247 Barrer and CO_2_/N_2_ selectivity of 56.1, which are 75% and 59% higher than those of the pristine Pebax membrane, respectively. These results indicate that incorporating PIM-g-MOF particles in the MMMs significantly improves their gas separation performance, with an optimal value of 1 wt% loading.

The enhancement in CO_2_ permeability and CO_2_/N_2_ and CO_2_/CH_4_ selectivities is likely due to the presence of the PIM as CO_2_-philic and free-volume-enhanced moieties in the interlayer space of the UiO-66-NH_2_ particles. The presence of PIM functional groups on the surface of the MOF particles also leads to hydrogen bonding interactions between the hard segments of the polymer matrix and the fillers (as confirmed by FTIR, SEM, and AFM spectroscopy). The formation of these hydrogen bonds enhances the interfacial compatibility between the polymer chains and the PIM-g-MOF fillers, which are required for fast and selective gas separation. At higher PIM-g-MOF loadings, such as 3 wt%, the CO_2_ permeability decreased, but, notably, the selectivity of the gases remained higher than that of the pristine polymer matrix.

In all of the pristine polymers and MMMs, the gas permeability decreased in the order P(CO_2_) > P(CH_4_) > P(N_2_). This result is consistent with the typical penetration behavior observed in semicrystallinePebax-based polymers and porous MOF-based MMMs. In addition, as the MOF content increased, the permeability started to increase (Figure 7a and Appendix A). Furthermore, this order is consistent with the increasing order of the critical temperature of the gases, which is related to their condensability, as shown in Figure 7b. In contrast, the CO_2_ selectivity (over N_2_ and CH_4_) of the MMMs increased with increasing MOF content up to 1 wt%. The CO_2_/N_2_ and CO_2_/CH_4_ selectivity increased from 35.3 and 11.7 to 56.1 and 17.1, respectively (Figure 7c and Appendix A).

Generally, the molecular permeation through gasseparation membranes is determined by two crucial properties: solubility and diffusivity [49]. A detailed analysis of gas diffusivity and solubility is summarized in Figure 7d to further investigate the enhancement of the CO_2_ permeability of the PIM-g-MOF-x MMMs. The increase in CO_2_ permeability in PIM-g-MOF-1 is mainly attributed to the higher adsorption and diffusion coefficients for CO_2_, which follows the same order as permeability: Pebax < PIM-g-MOF-0.5 < PIM-g-MOF-1 (Figure 7d). Incorporation of highly rigid and porous MOFs into the polymer matrix by physical bonding causes disordered polymer chain packing and reduces the crystallinity of the Pebax polymer (shown by XRD analysis). As a result, the free volume for gas diffusion increases. Furthermore, the well-dispersed MOFs in the polymer matrix facilitate the diffusion of the gas molecules through its pore structure, as this diffusion path requires less energy (E_d_) than diffusion through the polymer matrix.

Along with the diffusivity, the MOFs also enhance the solubility of the gases (Figure 7d), which has also been reported in the literature [13]. Consequently, the gas permeability increases to a level where the MOF particles are well dispersed in the polymer matrix (up to 1 wt% MOF-loaded membrane, PIM-g-MOF-1) (Appendix A and Figure 7). This enhancement in permeability is significant, as many previous studies have reported no significant performance improvement for Pebax polymer membranes even with a high loading of filler (over 5–20 wt%) [46,50].

However, the permeability of the PIM-g-MOF-x starts to decrease at 3 wt% MOF loading (PIM-g-MOF-3) and decreases dramatically at 5% loading (PIM-g-MOF-5). The permeability of PIM-g-MOF-3 (196.5 Barrer) was still higher than that of the pristine Pebax membrane (Appendix A), and the permeability further decreased to ~114.3 Barrer for PIM-g-MOF-5, which is lower than that of the pristine Pebax polymer matrix. There might be several reasons for the reduced permeability of PIM-g-MOF-x at high MOF loading (PIM-g-MOF-5). These include aggregation of MOFs causing polymer densification at the interface, preferential interaction of the oxygen in the ether group with the metal ion in PIM-g-MOF causing reduced CO_2_ affinity of the polymer, and the aggregation of the MOFs hindering the passage of gases through their pore structures.

The aggregation and consequent densification of polymers surrounding the accumulated MOFs are analogous to wrapping a rigid material with an elastic one and are thought to cause a barrier to gas transport. This causes a reduction in the gas permeability of PIM-g-MOF-x compared to the pristine polymer. The diffusivities of the PIM-g-MOF-x followed the same order as the permeability (PIM-g-MOF-1 > PIM-g-MOF-3 > PIM-g-MOF-5), thus supporting the hypothesis of aggregation or polymer densification.

For comparison, gas separation was also carried out using the 1 wt% unmodified UiO-66-NH_2_-loaded blended membrane, UiO-66-NH_2_-1. The permeability for all gases of the UiO-66-NH_2_-1 membrane is quite low compared to that of PIM-g-MOF-0.5 and PIM-g-MOF-1. For example, the CO_2_ permeability (150.2 Barrer) of UiO-66-NH_2_-1 is 60% lower than that of the PIM-g-MOF-1. A high aggregation of pure MOFs in the MMM was found in SEM analysis and is believed to be the reason for the poor separation performance. The results from the blended membrane strongly indicate that the UiO-66-NH_2_ MOFs functionalized with PIM played a crucial role in the synthesis of defect-free MMMs with well-dispersed MOFs, which enhanced the gas separation performance.

The CO_2_ selectivity (over non-polar N_2_ and CH_4_) of the PIM-g-MOF-x is very high (50–56.1), and the selectivity slightly decreased with increased MOF content as follows: PIM-g-MOF-1 > PIM-g-MOF-3 > PIM-g-MOF-5 > pristine Pebax (Appendix A and Figure 7c). The changes in selectivity with increased permeability indicate that the MOF-loaded MMMs did not form the undesirable sieve-in-a-case interface morphology [13]. These results indicate that the size selectivity in the PIM-g-MOF-x originating from the Pebax copolymer is further improved by incorporating MOFs in the MMMs.

However, the selectivity (CO_2_/N_2_) of the unmodified MOF-containing membrane, UiO-66-NH_2_-1 (Appendix A), is far lower than that of all of the functionalized MOF-based MMMs, and the value is almost 102% lower than that of PIM-g-MOF-1 with the same amount of MOF loading (i.e., 1 wt%). This result is attributed to an excessive loss of diffusivity and, hence, solubility, which is most likely due to the formation of a non-ideal interface (a “plugged sieve”) between the polymer matrix and the unmodified UiO-66-NH_2_. Overall, both gas permeability and selectivity increased simultaneously in the MMMs up to an optimum level of MOF loading (~1 wt%), demonstrating a very promising material for gas separation applications.

### 3.7. Permeability vs. Selectivity of the PIM-g-MOF-x

The validity of the CO_2_/N_2_ separation performance of the PIM-g-MOF-x MMMs was verified by plotting the experimental results on the Robeson 2008 upper bound plot (Figure 8) [11]. Data from high-performance Pebax-based MMMs, various Pebax-based membranes, and other commercial polyimide-based MMMs are included for comparison. The pristine Pebax membrane has moderate permeability with good selectivity for CO_2_/N_2_. The incorporation of the PIM-g-MOF (**1**) fillers into Pebax membranes significantly improves both the gas permeability and selectivity. The results indicate that the developed MMMs could be a potential candidate for CO_2_ separation applications. The separation performance of CO_2_/N_2_ lies on the 2008 upper bound and is better than that of pristine Pebax [44,50,51,52], various Pebax-based MMMs [46,50,52,53,54,55,56], and commercial polymers, including CA [57,58], Matrimid [18,59], etc.

### 3.8. Anti-Aging Performance

The structural stability of the proposed MMM was investigated by carrying out a long-term operation test, as shown in Figure 9a. The best-performing membrane, PIM-g-MOF-1, was chosen to investigate the operatingtime effect on the gasseparation performance. During 240 days of testing, the PIM-g-MOF-1 MMM exhibited high and stable separation performance in terms of CO_2_ permeability and CO_2_/N_2_ selectivity, indicating that the membrane has excellent stability against acidic CO_2_ gas. Furthermore, the CO_2_/N_2_ selectivity slightly increased over time after 235 days. This swelling-induced densification was mainly attributed to the gradual reduction of free volume caused by rearrangement of the network structure [13]. Overall, the results indicated that stable CO_2_/N_2_ separation performance was obtained for the best-performing membrane—PIM-g-MOF-1 membranes.

### 3.9. Effect of Temperature on GasSeparation Performance

The temperature dependence of the gas permeability and selectivity was investigated using the best-performing membrane at 30 to 50 °C and 1.0 atm. The CO_2_ permeability gradually increased as the temperature increased, which is attributed to the improved mobility of the gas molecules and the flexibility of the Pebax chain (Figure 9b) [60]. Conversely, excessive swelling of the polymer reduced the CO_2_/N_2_ selectivity. Despite selectivity loss at high temperature, the CO_2_/N_2_ selectivity of the PIM-g-MOF-1 membrane was still high, at 29, with a permeability of 460 Barrer, showing the promise of the material even at high temperature.

### 3.10. Pressure Effect on Gas Separation

The variation of the permeability coefficient with pressure for PIM-g-MOF-1 (1 wt% MOF loading) membrane was analyzed at up to 20 atm, and the normalized results are presented in Appendix A. The permeability of CO_2_ was decreased initially at 1 atm, followed by a gradual increase of permeability for further increases of pressure up to 20 atm. However, the permeability of N_2_ in the PIM-g-MOF-1 membrane gradually decreased faster with increased pressure up to 20 atm, whereas the CO_2_ permeability increased. As a result, the CO_2_/N_2_ selectivity gradually increased with the pressure.

As mentioned, CO_2_ has higher condensability than N_2_, and so an increase in feed pressure results in a considerable improvement in sorption and solubility. The interaction of CO_2_ with the ether groups in Pebax due to dipole–quadrupole interactions also has an important role in CO_2_ permeation, but this interaction does not occur with non-polar N_2_ molecules. In the membranes prepared in this study, the solubility of the gases is the most effective parameter regarding their permeability. Overall, the effect of pressure on CO_2_/N_2_ separation is beneficial for PEO-containing membrane, as it enhances both permeability and selectivity at high pressure. Similar results have been reported previously [47,50,61].

## 4. Conclusions

This study has provided an effective strategy to prepare MMMs by embedding PIM-g-MOFs into a Pebax polymer matrix. The functionalized PIM-g-MOFs contained more organic components on their surfaces that improved the compatibility between Pebax and the MOFs. This contributed to improvements in CO_2_ permeability by 75% and CO_2_/N_2_ selectivity by 59% of the PIM-g-MOF-1 MMM compared to the pristine Pebax membrane. The PIM-g-MOF-1 (1 wt% PIM-g-MOF loading) membranes showed the best CO_2_ separation performance, which was on the 2008 upper bound.

The mechanical properties of the membranes were improved by the PIM-g-MOF, which improved the applicability of the membranes for real applications. Furthermore, the developed MMMs had excellent anti-aging performance for up to 240 days. The key factor of this study is that it contributes an excellent approach of MOF modification for the development of gas separation technology. Furthermore, it has provided valuable insight into the use of PIM-g-MOF incorporation in Pebax copolymer with a combined structure of hard and soft segments for effective gas separation membranes.

## Data Availability

Not applicable.

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
