# Peer review of "Mixed Matrix Membranes for Efficient CO2 Separation Using an Engineered UiO-66 MOF in a Pebax Polymer"

_polymers, 2022, doi:10.3390/polym14040655_

Round 1

Reviewer 1 Report

A paper by Husna et. al. reports a three-step method for the preparation of mixed matrix membrane combining PIM-ordered crystallites of UiO-66-NH2 and pebax polymer both comprising selective gas adsorption properties. The manuscript is well written and presents important results. Some short critical notes should be considered before acceptance: 

1. What is meant under stability in lines 227-238? The framework in UiO-66-NH2 is known to decompose at 350-400 °C, so the discussion about increasing thermal stability after PIM treatment is doubtful. Elemental analysis for MOF and MOF-PIM composite could help to assign weight loss steps more accurately.

2. DMF has a high boiling point and much larger molecular size than CO2/CH4/N2 so its evaporation from the composite at room temperature is doubtful again. Are there any proofs of full activation of MOF structure prior gas permeability measurements? 

3. Some more recently reported low-enthalpy selective CO2 adsorbents could be mentioned in the introduction [10.1021/acs.inorgchem.8b00138; 10.3390/molecules25194396; 10.1039/C8NJ00109J].

4. Some minor typos

line 63: molecule → atom

lines 74-75: high free volume in polymer

line 140: were → was 

etc

Reviewer 2 Report

The manuscript has some merits, it presents a composite membrane for gas separation. It  could be published in Polymers but more characterization is needed as well as other improvements per below:

1) Very similar works have been published in the literature. Clarify what is the novelty of the work, and how the field has been advanced.

2) PIM-1 synthesis requires TFTPN which has F as the leaving group, and an electron withdrawing group (CN) in the orto position. The authors used compound 4 with 2x Cl instead of 2x F and without the activating groups in orto position. In principle this reaction should not work. Evidence should be provided that 4 can react with 3.

3) BET surface area of the prepared compounds 1, 2 and 5 should be reported and compared. This is a crucial aspect of the presented research. Similarly, XPS characterization should be given in the manuscript.

4) Elemental microanalysis should be reported, in particular for the metal and the Cl content/absence.

5) Cross-sectional SEM images across the membrane, and on the top surface are essential. EDX should also be incorporated to show the MOF distribution.

6) Polymer attachment to UiO-66-NH2 for mixed matrix membranes was previously reported, which should be briefly acknowledged (10.1039/D1TA06205K).

7) UiO-66/Pebax composite has been used for gas separation, which should be mentioned and the gas separation performance compared with the new approach by the authors (10.1039/C7TA07512J).

8) The scale bars in Figure 6 are not legible. Enlarge them. They cannot be read.

9) Reproducibility is an important factor in science. It is necessary to present error bars for the data in Figure 7 and 9 as well as Figure 3b.

Round 2

Reviewer 2 Report

The auhtors worked on all the comments and the manuscript can be accepted now.